# Application of Imaging and Artificial Intelligence for Quality Monitoring of Stored Black Currant (*Ribes nigrum* L.)

**DOI:** 10.3390/foods11223589

**Published:** 2022-11-11

**Authors:** Ewa Ropelewska

**Affiliations:** Fruit and Vegetable Storage and Processing Department, The National Institute of Horticultural Research, Konstytucji 3 Maja 1/3, 96-100 Skierniewice, Poland; ewa.ropelewska@inhort.pl

**Keywords:** black currant storage, image processing, texture parameters, machine vision, classification

## Abstract

The objective of this study was to assess the influence of storage under different storage conditions on black currant quality in a non-destructive and inexpensive manner using image processing and artificial intelligence. Black currants were stored at a room temperature of 20 ± 1 °C and a temperature of 3 °C (refrigerator). The images of black currants directly after harvest and fruit stored for one and two weeks were obtained using a digital camera. Then, texture parameters were computed from the images converted to color channels *R* (red), *G* (green), *B* (blue), *L* (lightness component from black to white), *a* (green for negative and red for positive values), *b* (blue for negative and yellow for positive values), *X* (component with color information), *Y* (lightness), and *Z* (component with color information). Models for the classification of black currants were built using various machine learning algorithms based on selected textures for RGB, Lab, and XYZ color spaces. Models built using the IBk, multilayer perceptron, and multiclass classifier for textures from RGB color space, and the IBk algorithm for textures from Lab color space distinguished unstored black currants and samples stored in the room for one and two weeks with an average accuracy of 100%, and the kappa statistic and weighted averages of precision, recall, Matthews correlation coefficient (MCC), receiver operating characteristic (ROC) area, and precision–recall (PRC) area equal to 1.000. This indicated a very distinct change in the external structure of the fruit after the first week and more and more visible changes in quality with increasing storage time. A classification accuracy reaching 98.67% (multilayer perceptron, Lab color space) for the samples stored in the refrigerator may indicate smaller quality changes caused by storage at a low temperature. The approach combining image textures and artificial intelligence turned out to be promising to monitor the quality changes in black currants during storage.

## 1. Introduction

Black currant (*Ribes nigrum* L.) is a deciduous shrub with dark fruit native to northern and central Europe and Asia. Black currant is grown for the production of fruit and for ornamental purposes. The cultivation is a cost-effective and profitable mainly due to using black currant as a valuable ingredient in a healthy diet [1]. Black currant shrubs have high cold tolerance and can be cultivated in cooler and humid areas. In addition to crops for economic purposes, black currant shrubs are important components of forest ecosystems of the Northern Hemisphere in terms of their productivity and diversity [2]. After domestication, black currant was spread throughout Europe, China, North America, and New Zealand. Poland is considered the top producer and the primary exporter of fresh and processed black currant. Due to its importance in human nutrition and suitability for industrial processing, the cultivation of black currants is in continuous expansion [3,4].

Fruits are small, astringent berries containing bioactive compounds and nutrients with health benefits. Black currants contain polyphenols and high levels of calcium, potassium, and phosphorus [5]. Black currants are rich in flavonoids, anthocyanins, organic acids, vitamins, polysaccharides, and unsaturated fatty acids [6]. The unique flavor of black currant and its health-promoting and nutritional properties, mainly due to the high contents of phenolic acids, anthocyanins, and vitamin C, contributed to increasing its consumption [7,8,9]. Black currants are characterized by typical flavor and distinct purple-black color. Black currant crop is economically important and especially popular in northern and eastern European countries [10]. Black currants can be consumed fresh and in their processed forms, among others, as jams, juices, jellies, purees, yogurt ingredients, and syrups with health-promoting properties and nutraceutical composition [3,10]. Black currant fruit and buds can also be used for the production of alcoholic beverages, such as wine, spirits, or liqueur, and essential oil for perfumes and food flavoring [11]. Black currant, due to its high level of colored pigments, can be a natural food colorant and dye [12]. As a nutraceutical, black currant can regulate or alleviate some diseases [12]. Fruit can be characterized by antioxidant, antimicrobial, anticancer, anti-inflammatory, anti-obesity, and immuno-stimulating effects and can protect blood vessels, improve eye functions, and promote dark adaptation [6,13,14].

Black currants in their fresh form are available for a few weeks in the year. They are very perishable and have a very short shelf life. Therefore, lowering the storage temperature or even freezing them can extend their shelf life [15,16]. Black currant is a non-climacteric fruit. The respiration rate decreases after harvest and is significantly higher for freshly harvested fruit than for the air-stored samples. Black currants designated for consumption can be stored under room conditions for about 2–3 weeks [17]. Storage can cause some quality changes in black currant [18]. Generally, berries as non-climacteric fruit are harvested fully ripened and are characterized by relatively rapid water loss and susceptibility toward damage and decay. Changes in berry quality during storage can also include the content of polyphenols, vitamins, minerals, titratable acidity, soluble solids, and pH. Refrigeration and freezing used for short- and long-term storage protect against the loss of heat-sensitive nutrients. Lowering the storage temperature reduces respiration rate, mold growth, pigment degradation, chemical changes, enzymatic reactions, and changes in cell structure [19]. Besides chemical composition and internal structure, external characteristics of black currant berries are important quality parameters. For example, fruit color and gloss are essential quality indicators [20]. In addition to the quality and chemical composition of berries, sensory attributes related to taste, appearance, and structure can also change during storage [21,22].

Instead of traditional manual fruit quality evaluation, machine vision enables more accurate and rapid detection of wrinkled berries and other defects caused by water loss, mechanical damage, or fungal decay. Non-destructive fruit quality evaluation using computer vision techniques and artificial intelligence can reduce food postharvest losses [23]. Artificial intelligence (AI) is increasingly used in agriculture, including precision agriculture and smart farming, also to obtain better yield quality. Artificial intelligence can help humanity address some of the most important challenges, such as feeding a rapidly increasing human population [24]. AI techniques are widely used in various fields of smart farming and agriculture for monitoring the quality of fruit [25]. The robustness and fault tolerance of fruit quality control can be enhanced by the ability to learn and continuously adapt AI systems. Computer vision systems are composed of an imaging device, e.g., a digital camera; a light source; and computer hardware with software. These systems can enable the measurements of quality parameters of products due to image acquisition, segmentation, feature extraction, classification, and data interpretation [26]. Computer vision as a part of AI allows the classification of fruit based on quality and identification of damages [27]. Fruit classification using machine vision using appropriate image features and learning algorithms can result in achieving the maximum economic value of fruit efficiently and accurately [28].

In this study, an innovative approach to monitoring the quality of black currant during storage was used. The undertaken research was intended to extend the use of artificial intelligence in fruit quality assessment. This study aimed to prove the effectiveness of the models developed based on image textures using artificial intelligence algorithms to identify changes in the quality of black currant with increasing time during storage under various conditions. The novelty of the present study was related to the classification of black currant samples using models built for textures selected for color space RGB including images from color channels *R*, *G*, and *B*; color space Lab, images from color channels *L*, *a*, and *b*; and color space XYZ, images from color channels *X*, *Y*, and *Z*. In the case of each color space, textures with the highest power for distinguishing the samples were selected from sets of 543 image parameters. The application of image analysis and artificial intelligence allowed for the objective, non-destructive, fast, and inexpensive assessment of the quality of stored black currant. 

## 2. Materials and Methods

### 2.1. Materials

The black currants were harvested on 1 August from a garden located in Olsztyn in northeastern Poland. Black currant berries were characterized by maturity but were not overripe. The black currant bunches were transported to the laboratory in plastic boxes with perforated walls. Only fully developed and undamaged black currants from each bunch were considered. Storage experiments began immediately after the fruit was harvested. Two hundred black currants were sampled to divide them into a set of one hundred objects intended to be stored in a room and a set of one hundred objects for storage in a refrigerator. Berries were arranged in a single layer at the bottom of the plastic boxes with perforated walls. One hundred black currants stored in the room were placed in a shaded place at an ambient temperature of 20 ± 1 °C. The remaining one hundred black currants were stored at a low temperature of 3 °C (refrigerator). Black currants were stored for two weeks.

### 2.2. Image Processing

The vision system was composed of a digital camera (Canon Inc., Tokyo, Japan), two LED (light-emitting diode) lamps as a light source, and a computer (HP Inc., Palo Alto, CA, USA) with programs for image analysis and classification. The imaging was performed in a completely dark room. Black currants were placed on a white surface. Samples were imaged directly after harvest, treating the part for storage in the room and the part for storage in the refrigerator separately. For each set of one hundred black currants, five images were obtained with twenty fruits in each image. Then, samples were subjected to storage for one week. After one week, black currants stored at room temperature and in the refrigerator were imaged again. In the case of the fruit stored in the refrigerator, no changes in the shape and structure of the outer surface were visible to the naked eye. In the case of samples kept in the room, only some of the fruit showed visible quality changes, such as wrinkling, water and mass losses, color fading, and changes in gloss. Based on visual observations, it was decided to extend the storage for another week. Afterward, the images of samples stored in the room and the refrigerator for two weeks were acquired. After two weeks of storage under room conditions, all black currants were wrinkled with visible loss of water and mass and changes in color and gloss that indicated a distinct change in quality. The changes were less noticeable for the fruit stored at a lower temperature in the refrigerator. However, after two weeks of storage, the quality of the fruit in the room did not allow the experiment to be continued. 

The obtained images for unstored black currant and samples stored for one and two weeks in the room and the refrigerator were processed with the use of the MaZda software (Łódź University of Technology, Institute of Electronics, Łódź, Poland) [29,30,31]. Firstly, the images were converted to the BMP (bitmap) file format, and the background of the images was changed to black (intensity of 0). Then, the image conversion to individual color channels *R*, *G*, *B*, *L*, *a*, *b*, *X*, *Y*, and *Z*; the image segmentation based on the pixel brightness intensity; and image texture extraction were carried out using MaZda (Łódź University of Technology, Institute of Electronics, Łódź, Poland). In the case of each fruit, 181 texture parameters based on the gradient map, histogram, autoregressive model, co-occurrence matrix, and run-length matrix were computed for images from each color channel.

### 2.3. Classification of Black Currant Stored under Different Conditions

The dataset was divided into six classes: black currants at the beginning of storage in the room (room 0), black currants stored at room temperature for 1 week (room 1) and 2 weeks (room 2), black currants before storage in the refrigerator (refrigerator 0), and black currants after storage in the refrigerator for 1 week (refrigerator 1) and 2 weeks (refrigerator 2). The classification was performed to compare the influence of prolonged time of storage at room temperature and a lower temperature in the refrigerator on black currant quality. Models for distinguishing black currants were built for classes of room 0 vs. room 1 vs. room 2 and refrigerator 0 vs. refrigerator 1 vs. refrigerator 2. The models were developed using WEKA machine learning software (Machine Learning Group, University of Waikato, New Zealand) [32,33,34] based on selected textures for RGB, Lab, and XYZ color spaces. The texture selection was carried out using the best-first search algorithm with the correlation-based feature selection (CFS) subset evaluator. A test mode of 10-fold cross-validation was used for the model development. The algorithms from groups of Lazy, Functions, Meta, Rules, Trees, and Bayes were tested, and the results obtained using one most effective algorithm from each group were chosen to be presented in this paper. The results obtained for each model are shown as average accuracy; kappa statistic; and weighted averages of precision, recall, Matthews correlation coefficient (MCC), receiver operating characteristic (ROC) area, and precision–recall (PRC) area [35,36]. For selected models, confusion matrices are also presented to better understand the impact of storage on black currant samples.

## 3. Results and Discussion

The black currant samples were classified as unstored vs. stored in the room for one week vs. stored in the room for two weeks and unstored vs. stored in the refrigerator for one week vs. stored in the refrigerator for two weeks. The exemplary images of unstored black currants and samples stored under different conditions for different times are presented in Figure 1.

Among the algorithms belonging to different groups, IBk from the group of Lazy, multilayer perceptron from Functions, multiclass classifier from Meta, JRip from Rules, random forest from Trees, and Bayes net from Bayes turned out to be the most effective. In the first stage of the classification of black currant samples, models were developed based on textures selected from sets of combined features from *R*, *G*, and *B* color channels of images belonging to the color space RGB. In the case of storage at room temperature, the unstored sample and black currants stored for one and two weeks were distinguished with an average accuracy reaching 100% for models developed using the IBk, multilayer perceptron, and multiclass classifier (Table 1). The kappa statistic and weighted averages of precision, recall, MCC, ROC area, and PRC area were equal to 1.000. This meant that all three classes were completely different in terms of selected image textures of the outer surface of black currant and were distinguished with 100% correctness from other samples. Thus, the strong influence of storage time on the fruit quality was revealed. The structure of the black currant surface changed after one week of storage sufficiently to distinguish the unstored sample from those stored for a week with 100% accuracy. Storage for another week influenced further changes in the quality of black currants. Classes of samples stored for two weeks and one week were correctly classified in 100% of cases and fruit samples stored for two weeks were also distinguished from unstored ones with an accuracy of 100% (Figure 2a). Due to the differences in quality of all three samples (unstored, stored in the room for one and two weeks) allowing for their completely correct differentiation, the average accuracy was equal to 100%. Models developed using other machine learning algorithms provided also satisfactory results including an average accuracy of up to 99.32% for random forest and Bayes net and 98.67% for JRip (Table 1). For the JRip algorithm providing the lowest average accuracy, samples stored in the room for one and two weeks were correctly distinguished from each other and unstored class in 100% of cases. Unstored black currants were correctly classified in 96% of cases, and the remaining 2% were included in the class of fruit stored for one week and 2% in the class of sample stored for two weeks (Figure 2b).

In the case of models built based on selected textures for RGB color space to classify unstored black currants and samples stored in the refrigerator for one and two weeks, a 100% accuracy was not achieved (Table 2). This may indicate a reduced negative influence of storage at lower temperature on the quality of black currants. All classes were distinguished with an average accuracy of up to 96.67% for models built using IBk, multilayer perceptron, and Bayes net. For these algorithms, the kappa statistic reached 0.950 and recall reached 0.967. The values (weighted averages) of precision equal to 0.968 and MCC equal to 0.951 were the highest for IBk and multilayer perceptron. The ROC area of 0.998 and PRC area of 0.996 were the highest for Bayes net (Table 2). Different values of other performance metrics, despite the same average accuracy, resulted from different accuracies of classifying individual samples as shown in Figure 3a,b. For example, in the case of the IBk algorithm, black currants stored in the refrigerator for two weeks were completely correctly classified (100%) as stored in the refrigerator for two weeks, and for Bayes net, the correctness was equal to 96%. The sample stored in the refrigerator for one week was classified with accuracies of 92% for IBk and 96% for Bayes net, whereas 98% of black currants belonging to the actual class unstored were correctly classified as unstored for both IBk and Bayes net algorithms. The JRip algorithm was characterized by the lowest average accuracy (89.33%) in distinguishing unstored fruit and samples stored in the refrigerator for one and two weeks. In addition, other metrics were the lowest and equal to 0.840 for kappa statistic, 0.896 for precision, 0.893 for recall, 0.841 for MCC, 0.932 for ROC area, and 0.857 for PRC area (Table 2). Low values of classification performance metrics for the JRip algorithm were related to the low correctness of distinguishing between individual classes. The accuracies of classification of the unstored sample and black currants stored for one and two weeks were equal to 92%, 86%, and 90%, respectively (Figure 3c). 

For the Lab color space, the obtained results were also very high. However, an average accuracy of 100% and the kappa statistic and weighted averages of precision, recall, MCC, ROC area, and PRC area equal to 1.000 for distinguishing unstored black currants and samples stored in the room for one and two weeks were found only for a model built using the IBk algorithm (Table 3). In the case of results for storage at room temperature, other algorithms allowed the classification of samples with an average accuracy of 96.67% for JRip to 99.33% for multilayer perceptron. Confusion matrices shown in Figure 4 confirmed the completely correct distinguishing of three samples for IBk (Figure 4a) and revealed the lowest accuracies equal to 92% for the unstored sample, 98% for the sample stored in the room for one week, and 100% for fruit stored in the room for two weeks in the case of JRip (Figure 4b).

The completely correct classification of three samples, namely unstored and stored in the refrigerator for one and two weeks, was not observed for any model built based on selected textures from the Lab color space (Table 4). The average accuracies were lower, from 91.39% for JRip to 98.67% for multilayer perceptron. The model developed using multilayer perceptron provided the highest values of kappa statistic (0.9801), precision (0.987), recall (0.987), MCC (0.980), ROC area (1.000), and PRC area (0.999). The accuracies for individual classes were the highest for multilayer perceptron (100% for the unstored sample and 98% for both samples stored for one and two weeks) (Figure 5a) and the lowest for JRip (96% for the unstored sample, 86% for the sample stored for one week, and 93% for the sample stored for weeks) (Figure 5b).

Models built based on selected textures from XYZ color space for the classification of stored in the room and unstored samples were characterized by the correctness from 93.33% for multiclass classifier to 98.67% for random forest (Table 5). The kappa statistic reached 0.980 for random forest. The weighted averages of precision (0.987), recall (0.987), and MCC (0.980) were also the highest for the model developed using random forest. The weighted averages of ROC area (0.999) and PRC area (0.998) were the highest for the model built using Bayes net classifying samples with an average accuracy of 97.33%. The confusion matrix obtained for the classification performed using random forest revealed an accuracy of 100% for black currants stored at room temperature for one week and 95% for the unstored sample and fruit stored in the room for two weeks (Figure 6a). Multiclass classifier, distinguishing classes with the lowest average accuracy, was characterized by 90% correctness for the unstored sample, 94% for fruit stored for one week, and 96% for black currants stored for two weeks in the room (Figure 6b).

The average accuracy of classification of unstored black currants and fruit stored in the refrigerator for one and two weeks reached 97.33% for models developed using random forest based on textures selected from XYZ color space (Table 6). For this model, other classification performance metrics were also the highest. The kappa statistic was equal to 0.9603. Additionally, the highest weighted averages of precision (0.974), recall (0.973), MCC (0.960), ROC area (0.998), and PRC area (0.996) were determined. The obtained confusion matrix showed that unstored fruit and sample stored in the refrigerator for two weeks were classified with an accuracy of 98%, and the sample stored in the refrigerator for one week were distinguished from others at an accuracy of 96% (Figure 7a). The lowest average accuracy was equal to 89.33% in the case of the multiclass classifier (Table 6). The multiclass classifier produced accuracies for individual classes equal to 98% for unstored fruit, 82% for fruit stored in the refrigerator for one week, and 88% for black currants stored in the refrigerator for two weeks (Figure 7b).

The combination of image analysis and artificial intelligence proved to be effective for monitoring the changes in black currants during storage. The experiments carried out included different storage conditions (at room temperature and in the refrigerator at 4 °C) and lengths of time (one week and two weeks). However, in view of the usefulness of the procedure used, there are many different possibilities to extend the scope of the experiments to other storage technologies that may also result in an extension of the storage time. Among others, it was reported that lowering the temperature to 0–1 °C combined with a relative humidity of 95% can be optimal conditions in a normal atmosphere (NA) for storing berries for up to 3 weeks. Furthermore, currants do not produce ethylene and are not susceptible to ethylene, but sulfur dioxide can be positively related to a modified atmosphere strategy [37]. In the case of the treatment of berries with sulfur dioxide during storage, abscission can be inhibited by the use of edible coatings [38]. Besides modified atmosphere packaging (MAP) and controlled atmosphere (CA), a successful solution for the storage of perishable fruit such as berries is the use of gaseous ozone (O_3_), which can be safer than other gases such as CO_2_, O_2_, and N_2_ [39]. Gaseous ozone can also be effective in the reduction of pesticide residue levels in currants and improvement of microbiological purity and can lead to berry fruit preservation and long-term storage [40,41]. Gaseous ozone can be applied as a pretreatment before the use of other storage techniques for berries, e.g., modified atmosphere packaging [42]. The application of ozone may result in improving the health-promoting properties of the fruit. However, the effect depends on the type and variety of fruit, the form and method of treatment, and the dosage of ozone [43]. The influence of the above-mentioned and other technologies on the external structure of black currants could also be assessed using image analysis and machine learning. This could allow the development of models to predict the maintenance of optimal fruit structure during storage under various conditions. Furthermore, different storage technologies can affect changes in biochemical and other quality indicators, e.g., total soluble solids, citric acid, vitamin C, the total content of phenolic compounds, color, firmness, and taste [22]. In further studies, models to predict the chemical properties of stored black currants can be developed. The available literature data indicate the possibility of using image processing and machine learning to estimate the chemical properties of plant materials [44,45].

## 4. Conclusions

Procedures for the classification of unstored black currants and fruit stored in the room for one and two weeks and in the refrigerator for one and two weeks were developed. The innovative models including textures selected separately for images from color space RGB, color space Lab, and color space XYZ were built using machine learning algorithms belonging to groups of Lazy, Functions, Meta, Rules, Trees, and Bayes. Combining image processing and artificial intelligence was a non-destructive and objective approach to quality monitoring of stored black currant. Due to the use of a digital camera to obtain the color images of unstored and stored black currant samples, image acquisition was performed quickly and inexpensively. The models developed proved to be very effective. The greater effect of storage in the room on the black currant quality was determined. The average accuracy of the classification of unstored samples and fruit stored at room temperature for one and two weeks reached 100%, whereas unstored black currants and samples stored in a refrigerator were distinguished with an average accuracy of up to 98.67%. This confirmed the possibility of determining the changes in black currant quality during storage using image features. Due to the promising results, further research to investigate the influence of other storage technologies on the structure of black currants and predict the chemical properties of stored fruit using image processing and machine learning may be undertaken.

## Figures and Tables

**Figure 1 foods-11-03589-f001:**
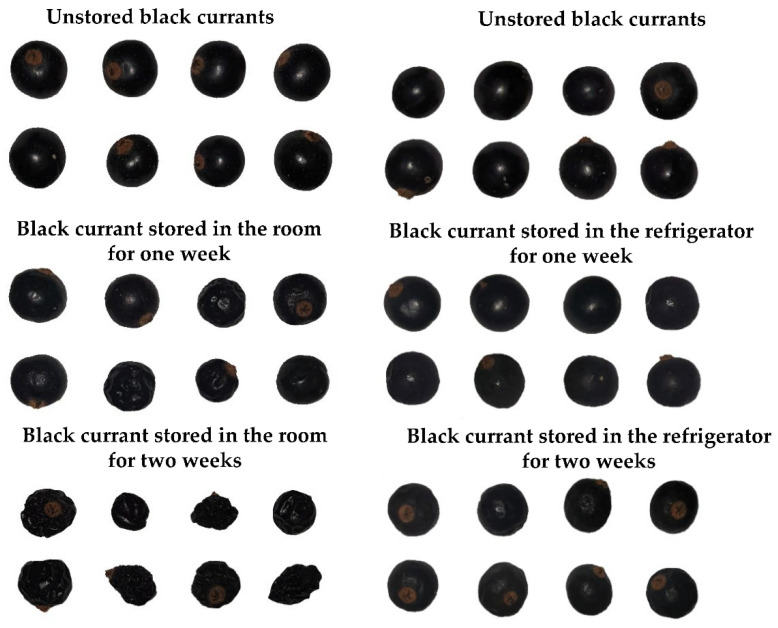
The images of unstored black currants at the beginning of experiments and samples stored in the room and the refrigerator for one week and two weeks.

**Figure 2 foods-11-03589-f002:**
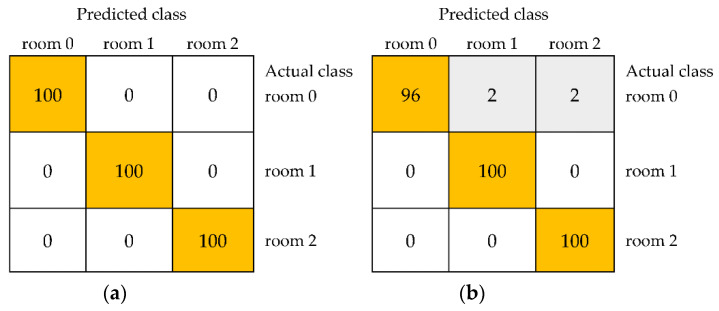
The confusion matrices of the classification of three classes of black currants, namely unstored (room 0) and stored at room temperature for 1 week (room 1) and 2 weeks (room 2), based on selected image textures from color space RGB using models developed using the IBk (**a**) and JRip algorithms (**b**).

**Figure 3 foods-11-03589-f003:**
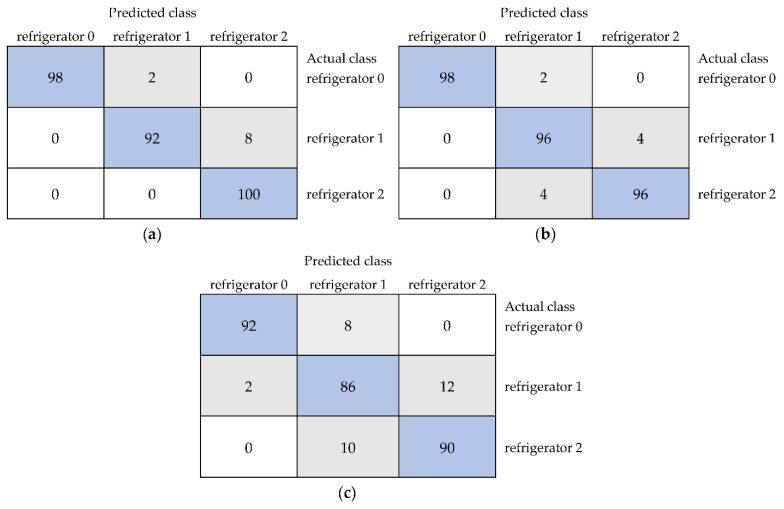
The confusion matrices of the classification of black currant samples, namely unstored (refrigerator 0) and stored in the refrigerator for 1 week (refrigerator 1) and 2 weeks (refrigerator 2), using selected image textures from color space RGB for the models built using the IBk (**a**), Bayes net, (**b**) and JRip algorithms (**c**).

**Figure 4 foods-11-03589-f004:**
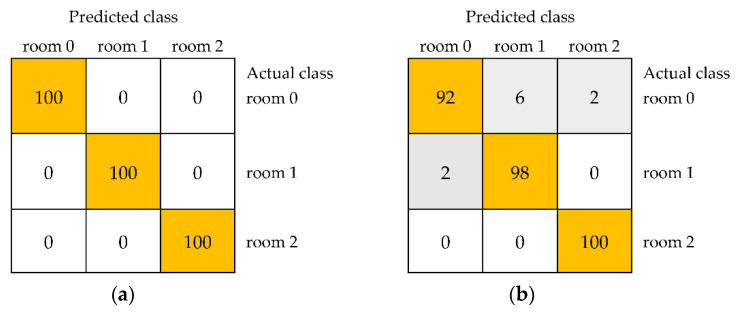
The confusion matrices of distinguishing unstored black currants (room 0) and fruit stored in the room for 1 week (room 1) and 2 weeks (room 2) using models including selected image textures from color space Lab built using the IBk (**a**) and JRip (**b**).

**Figure 5 foods-11-03589-f005:**
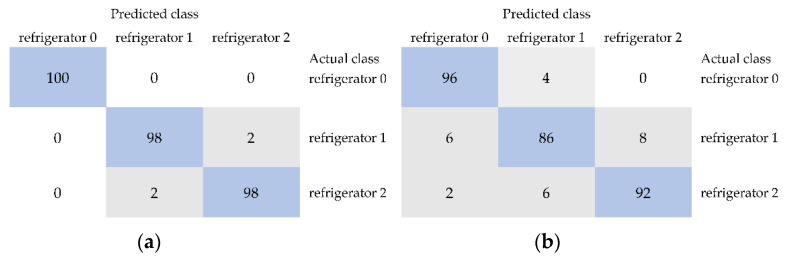
The confusion matrices of distinguishing unstored black currants (refrigerator 0) and samples stored in the refrigerator for 1 week (refrigerator 1) and 2 weeks (refrigerator 2) for models including selected image textures from color space Lab developed using the multilayer perceptron (**a**) and JRip (**b**).

**Figure 6 foods-11-03589-f006:**
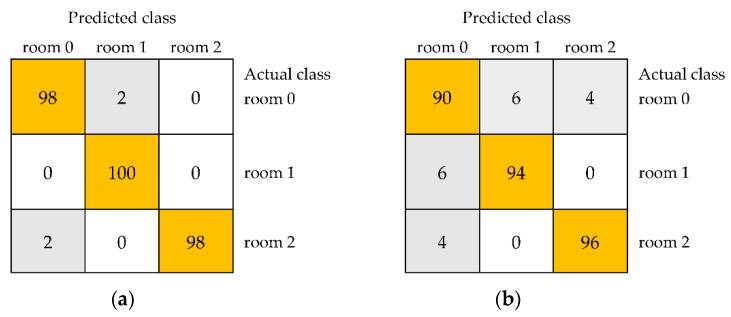
The confusion matrices of distinguishing unstored black currants (room 0) and samples stored in the room for 1 week (room 1) and 2 weeks (room 2) using models combining selected image textures from color space XYZ developed using random forest (**a**) and multiclass classifier (**b**).

**Figure 7 foods-11-03589-f007:**
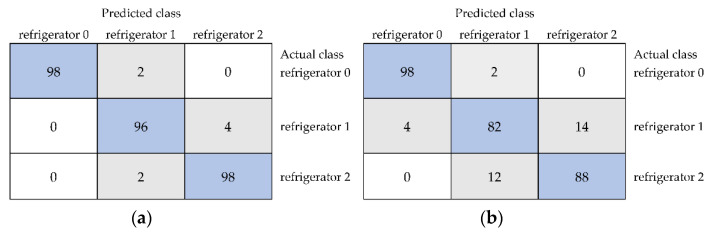
The confusion matrices of the classification of unstored black currants (refrigerator 0) and fruit stored in the refrigerator for 1 week (refrigerator 1) and 2 weeks (refrigerator 2) using models including selected image textures from color space XYZ built using random forest (**a**) and multiclass classifier (**b**).

**Table 1 foods-11-03589-t001:** The performance metrics of the classification of three classes of black currants (unstored, stored in the room for one week, and stored in the room for two weeks) based on models including selected image textures from color space RGB built using different algorithms.

Algorithm	Average Accuracy (%)	Kappa Statistic	Precision (Weighted Average)	Recall (Weighted Average)	MCC (Weighted Average)	ROC Area (Weighted Average)	PRC Area (Weighted Average)
lazy.IBk	100	1.000	1.000	1.000	1.000	1.000	1.000
functions.MultilayerPerceptron	100	1.000	1.000	1.000	1.000	1.000	1.000
meta.MultiClassClassifier	100	1.000	1.000	1.000	1.000	1.000	1.000
rules.JRip	98.67	0.9796	0.987	0.987	0.980	0.989	0.976
trees.RandomForest	99.32	0.9898	0.993	0.993	0.990	1.000	1.000
bayes.BayesNet	99.32	0.9898	0.993	0.993	0.990	1.000	1.000

MCC—Matthews correlation coefficient; ROC Area—receiver operating characteristic area; PRC Area—precision–recall area.

**Table 2 foods-11-03589-t002:** The results of the classification of black currants (unstored, stored in the refrigerator for one week and two weeks) using models developed based on selected image textures from color space RGB using algorithms from different groups.

Algorithm	Average Accuracy (%)	Kappa Statistic	Precision (Weighted Average)	Recall (Weighted Average)	MCC (Weighted Average)	ROC Area (Weighted Average)	PRC Area (Weighted Average)
lazy.IBk	96.67	0.950	0.968	0.967	0.951	0.975	0.947
functions.MultilayerPerceptron	96.67	0.950	0.968	0.967	0.951	0.993	0.989
meta.MultiClassClassifier	95.33	0.930	0.954	0.953	0.930	0.985	0.970
rules.JRip	89.33	0.840	0.893	0.893	0.841	0.932	0.857
trees.RandomForest	96.00	0.940	0.960	0.960	0.940	0.997	0.993
bayes.BayesNet	96.67	0.950	0.967	0.967	0.950	0.998	0.996

MCC—Matthews correlation coefficient; ROC Area—receiver operating characteristic area; PRC Area—precision–recall area.

**Table 3 foods-11-03589-t003:** The average accuracies, kappa statistic values, and weighted averages of other classification performance metrics for distinguishing unstored black currants and samples stored at room temperature for one week and two weeks using models combining selected textures from color space Lab.

Algorithm	Average Accuracy (%)	Kappa Statistic	Precision (Weighted Average)	Recall (Weighted Average)	MCC (Weighted Average)	ROC Area (Weighted Average)	PRC Area (Weighted Average)
lazy.IBk	100	1.000	1.000	1.000	1.000	1.000	1.000
functions.MultilayerPerceptron	99.33	0.990	0.993	0.993	0.990	0.998	0.997
meta.MultiClassClassifier	98.67	0.980	0.987	0.987	0.980	0.998	0.998
rules.JRip	96.67	0.950	0.967	0.967	0.950	0.977	0.955
trees.RandomForest	97.33	0.960	0.973	0.973	0.960	0.999	0.998
bayes.BayesNet	97.33	0.960	0.974	0.973	0.960	0.998	0.997

MCC—Matthews correlation coefficient; ROC Area—receiver operating characteristic area; PRC Area—precision–recall area.

**Table 4 foods-11-03589-t004:** The performance metrics of distinguishing unstored black currants and black currants stored in the refrigerator (for one week and two weeks) based on models including selected textures from color space Lab developed using different algorithms.

Algorithm	Average Accuracy (%)	Kappa Statistic	Precision (Weighted Average)	Recall (Weighted Average)	MCC (Weighted Average)	ROC Area (Weighted Average)	PRC Area (Weighted Average)
lazy.IBk	96.69	0.9503	0.968	0.967	0.951	0.973	0.944
functions.MultilayerPerceptron	98.67	0.9801	0.987	0.987	0.980	1.000	0.999
meta.MultiClassClassifier	98.01	0.9702	0.981	0.980	0.971	0.999	0.997
rules.JRip	91.39	0.8708	0.914	0.914	0.871	0.940	0.883
trees.RandomForest	96.69	0.9503	0.968	0.967	0.951	0.997	0.994
bayes.BayesNet	95.36	0.9305	0.954	0.954	0.933	0.996	0.993

MCC—Matthews correlation coefficient; ROC Area—receiver operating characteristic area; PRC Area—precision–recall area.

**Table 5 foods-11-03589-t005:** The results of distinguishing unstored black currants and samples stored in the room for one week and two weeks using models built based on selected textures from color space XYZ.

Algorithm	Average Accuracy (%)	Kappa Statistic	Precision (Weighted Average)	Recall (Weighted Average)	MCC (Weighted Average)	ROC Area (Weighted Average)	PRC Area (Weighted Average)
lazy.IBk	98	0.970	0.980	0.980	0.970	0.985	0.967
functions.MultilayerPerceptron	98	0.970	0.980	0.980	0.970	0.995	0.995
meta.MultiClassClassifier	93.33	0.900	0.933	0.933	0.900	0.969	0.954
rules.JRip	98	0.970	0.980	0.980	0.970	0.988	0.973
trees.RandomForest	98.67	0.980	0.987	0.987	0.980	0.997	0.996
bayes.BayesNet	97.33	0.960	0.973	0.973	0.960	0.999	0.998

MCC—Matthews correlation coefficient; ROC Area—receiver operating characteristic area; PRC Area—precision–recall area.

**Table 6 foods-11-03589-t006:** The performance metrics of the classification of unstored black currants and black currants stored in the refrigerator (one and two weeks) based on models developed based on selected textures from color space XYZ using different algorithms.

Algorithm	Average Accuracy (%)	Kappa Statistic	Precision (Weighted Average)	Recall (Weighted Average)	MCC (Weighted Average)	ROC Area (Weighted Average)	PRC Area (Weighted Average)
lazy.IBk	96.03	0.9404	0.961	0.960	0.941	0.967	0.933
functions.MultilayerPerceptron	94.70	0.9205	0.947	0.947	0.921	0.996	0.993
meta.MultiClassClassifier	89.33	0.841	0.893	0.893	0.841	0.983	0.964
rules.JRip	93.38	0.9006	0.935	0.934	0.901	0.961	0.930
trees.RandomForest	97.33	0.9603	0.974	0.973	0.960	0.998	0.996
bayes.BayesNet	95.33	0.930	0.954	0.953	0.930	0.991	0.984

MCC—Matthews correlation coefficient; ROC Area—receiver operating characteristic area; PRC Area—precision–recall area.

## Data Availability

The data presented in this study are available on request from the corresponding author.

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
