# Peer review of "Application of Imaging and Artificial Intelligence for Quality Monitoring of Stored Black Currant (Ribes nigrum L.)"

_foods, 2022, doi:10.3390/foods11223589_

Round 1
Reviewer 1 Report
Generally the article is well designed, but there are some issues which are ought to be taken into the consideration:
1. In the introduction section there must be added some information concerning the blackberries quality (Flavor, taste, content of sugars, vitamins etc.) changing during the storage period.
2. In the Materials and Methods section there was not any information concerning the harvesting period of berries. In addition there is only two-week storage period is taken into the consideration and not any information about the longer period.
3. Fig. 1 and the related information should be removed to the Results section.
4. In the Results section there should be included also some additional data of biochemical parameters of berries in the creation of the suggested digital model of evaluation of storage time.
Author Response
Generally the article is well designed, but there are some issues which are ought to be taken into the consideration:
- In the introduction section there must be added some information concerning the blackberries quality (Flavor, taste, content of sugars, vitamins etc.) changing during the storage period.
Response: Thank you very much for this comment. It is mentioned as follows: “Storage can cause some quality changes in black currant [18]. Generally, berries as non-climacteric fruit are harvested fully ripened and are characterized by relatively rapid water loss and susceptibility toward damage and decay. Changes in berry quality during storage can also include the content of polyphenols, vitamins, minerals, titratable acidity, soluble solids, and pH. Refrigeration and freezing used for short and long-term storage protect against the loss of heat-sensitive nutrients. Lowering the storage temperature reduces respiration rate, mold growth, pigment degradation, chemical changes, enzymatic reactions, the changes in cell structure [19]. Besides chemical composition and internal structure, external characteristics of black currant berries are important quality parameters. For example, fruit color and gloss are essential quality indicators [20]. In addition to the quality and chemical composition of berries, also sensory attributes related to taste, appearance, and structure can change during storage [21,22].”
- In the Materials and Methods section there was not any information concerning the harvesting period of berries. In addition there is only two-week storage period is taken into the consideration and not any information about the longer period.
Response: It has been indicated in the Materials and Methods as follows
“The black currants were harvested on August 1 from the garden located in Olsztyn in north-eastern Poland. Black currant berries were characterized by maturity but were not overripe. The black currant bunches were transported to the laboratory in plastic boxes with perforated walls. Only fully developed and undamaged black currants from each bunch were considered. Storage experiments began immediately after harvesting the fruit.”
“After one week, black currants stored at room temperature and in the refrigerator were imaged again. In the case of the fruit stored in the refrigerator, no changes in the shape and structure of the outer surface were visible to the naked eye. In the case of samples kept in the room, only some of the fruit showed visible quality changes, such as wrinkling, water and mass losses, color fading and changes in gloss. Based on visual observations, it was decided to extend the storage for another week. Afterward, the images of samples stored in the room and the refrigerator for two weeks were acquired. After two weeks of storage under room conditions, all black currants were wrinkled with visible loss of water and mass and changes in color and gloss that indicated a distinct change in quality. The changes were less noticeable for the fruit stored at a lower temperature in the refrigerator. However, after two weeks of storage, the quality of the fruit in the room did not allow the experiment to be continued.”
Additionally, there is the following information in the Introduction “Black currant is a non-climacteric fruit. The respiration rate decreased after harvest and is significantly higher for freshly harvested fruit than for the air-stored samples. Black currants designated for consumption can be stored under room conditions for about 2–3 weeks [17].”
- Harb, J.; Bisharat R.; Streif J. Changes in volatile constituents of blackcurrants (Ribes nigrum L. cv. ‘Titania’) following controlled atmosphere storage. Postharvest Biology and Technology 2008, 47, 271–279
- Fig. 1 and the related information should be removed to the Results section.
Response: It has been done. Figure 1 and its description are in the Results section.
- In the Results section there should be included also some additional data of biochemical parameters of berries in the creation of the suggested digital model of evaluation of storage time.
Response: The proposed procedure focused only on the use of a non-destructive and objective approach to quality monitoring of stored black currant. Therefore, combining image processing and artificial intelligence was chosen without destructive biochemical analysis. It was indicated that due to the use of a digital camera to obtain the color images of unstored and stored black currant samples, image acquisition was performed quickly and inexpensively. However, directions for further research have been indicated:
“Furthermore, different storage technologies can affect changes in biochemical and other quality indicators, e.g., total soluble solids, citric acid, vitamin C, the total content of phenolic compounds, color, firmness, and taste [22]. In further studies, models to predict the chemical properties of stored black currants can be developed. The available literature data indicated the possibility of using image processing and machine learning to estimate the chemical properties of plant materials [44,45].”
Reviewer 2 Report
The draft manuscript that has been submitted shows that a great deal of time and effort has been put into its preparation. I believe it deserves to be published in this journal, although I suggest some revisions to be considered. Most of my comments are of an editorial nature and should be considered to improve the paper's readability.
I have attached my comments to the file. Besides, I have corrected minute grammatical errors in it. Thanks

Author Response
Thank you very much for reading and reviewing the manuscript carefully. All your comments have been considered. The responses are in the attached file.

Round 2
Reviewer 1 Report
The article can be published .